# Dynamically Masked Discriminator for GANs

**Wentian Zhang**[1,2†]   **Haozhe Liu**[1,2†]   **Bing Li**[1✉⋆]   **Jinheng Xie**[3]   **Yawen Huang**[2]
**Yuexiang Li**[2,4✉]   **Yefeng Zheng**[2]   **Bernard Ghanem**[1]

[1] AI Initiative, King Abdullah University of Science and Technology
[2] Jarvis Research Center, Tencent YouTu Lab   [3] National University of Singapore
[4] Life Sciences Institute, Guangxi Medical University
`bing.li@kaust.edu.sa; leeyuexiang@163.com`

## Abstract

Training Generative Adversarial Networks (GANs) remains a challenging problem. The discriminator trains the generator by learning the distribution of real/generated data. However, the distribution of generated data changes throughout the training process, which is difficult for the discriminator to learn. In this paper, we propose a novel method for GANs from the viewpoint of online continual learning. We observe that the discriminator model, trained on historically generated data, often slows down its adaptation to the changes in the new arrival generated data, which accordingly decreases the quality of generated results. By treating the generated data in training as a stream, we propose to detect whether the discriminator slows down the learning of new knowledge in generated data. Therefore, we can explicitly enforce the discriminator to learn new knowledge fast. Particularly, we propose a new discriminator, which automatically detects its retardation and then dynamically masks its features, such that the discriminator can adaptively learn the temporally-vary distribution of generated data. Experimental results show our method outperforms the state-of-the-art approaches.

## 1 Introduction

Generative Adversarial Networks (GANs) [19, 84, 20, 5, 72, 70, 71] have shown the remarkable performance in various applications [89, 85, 81, 3, 44], which have attracted intensive interests. The main components of GANs are the generator and the discriminator, where the generator is trained to generate realistic samples, and the discriminator learns to distinguish real and generated samples. However, it is well-known that GANs are difficult to train [33, 30, 21].

Many efforts have been devoted to alleviating training difficulties for GANs. Some studies balance the generator and discriminator from the side of network architecture, such as DCGAN [61], PG-GAN [32], Style-GANs [35, 36, 34], and BigGAN [6]. Besides, some studies [2, 94, 53, 86] address this challenge from the learning objective, *e.g*., WGAN [2], EBGAN [94], LSGAN[53] and realness-GAN [86]. These methods can further stabilize training, especially coupled with the Lipschitz regularization [56, 21] and conditional information [55]. Recent works [87, 33, 30, 95, 93, 50, 48, 82] propose to improve GANs on the discriminator side. For example, different data augmentation strategies [33, 30, 95] are proposed to strengthen the discriminator for a limited data regime. Regularizations are applied to stabilize the training of discriminator [93, 50, 82] and combat the mode

---

† Equal Contribution. Haozhe worked on this project before joining KAUST.
✉ Corresponding Author
⋆ Project Leader
Code is available at `https://github.com/WentianZhang-ML/DMD`.

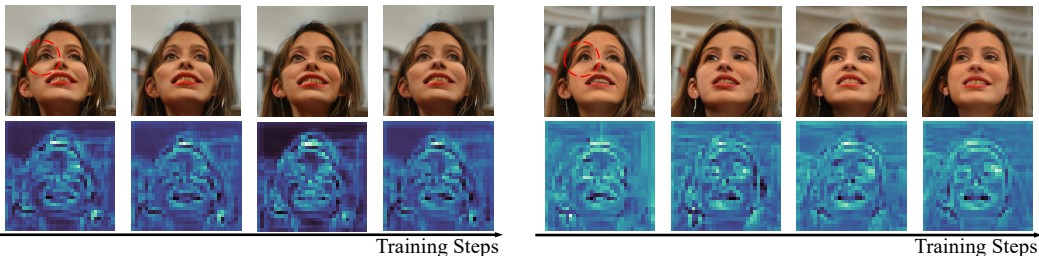

| (a) StyleGAN-V2 | (b) StyleGAN-V2 + Ours |

Figure 1: **Illustration of the advantage of our method**, where the first and second rows show generated images and feature maps taken from a discriminator layer, respectively. (a) StyleGAN-V2 introduces similar artifacts into generated images, despite the increased training steps. Our method is simple yet effectively improves the training of GANs, boosting the quality of generated samples for StyleGAN-V2 in (b). By fast learning incoming generated samples, the proposed discriminator captures artifacts in the local regions, better guiding the training of the generator. The averaging cosine similarity between the current and previous feature maps is 0.8353 for StyleGAN-V2 and 0.4232 for ours, which indicates that our method enforces the attentive regions of the discriminator to be more different than the baseline, better adapting to time-varying distribution of generated data.

collapse [48]. These methods show that the discriminator is very crucial to the training of GANs. Nevertheless, stably training GANs remains a challenging problem [33, 31, 93].

In this paper, we rethink the difficulty of training GANs from the discriminator side. The discriminator guides the training of the generator via learning the distributions of real/generated samples. However, since it has been observed that the characteristics of generated samples vary during the whole training process [87], a question arises: *how challenging is it for the discriminator to learn from such time-varying generated samples?* Our study shows that the distribution of generated samples dynamically changes during training, posing significant challenges to the discriminator. In particular, the discriminator in existing work (*e.g.*, [6, 36, 57, 50]) does not explicitly consider an online task that addresses time-varying distributions of generated samples but is presented for a typical classification task. We refer to such a discriminator as *fixed*. We find that the fixed discriminator often improperly relies on local features that are important to historically generated samples to distinguish incoming ones, although the distributions of these generated samples are different. As a result, the discriminator insufficiently learns the distribution of incoming generated samples, leading to subpar guidance to the generator (see Fig. 1).

We propose a novel method for training GANs, from the perspective of online continual learning. Online continual learning has attracted increasing attention [52, 14, 40]. Different from offline continual learning focusing on the catastrophic forgetting problem [51, 38, 60], recent online continual learning methods [17, 7, 47, 22] pointed out that data with time-vary distribution requires learning methods to have the ability of fast learning and quick adaption. Motivated by these methods, we propose to detect whether the discriminator slows down the learning of new knowledge in generated data by treating generated samples throughout training as a stream of data. We then explicitly force the discriminator to learn new knowledge fast. To this end, we propose a simple yet effective scheme to detect the retardation of the discriminator and force it to learn fast. Instead of designing a sophisticated network architecture, we dynamically mask discriminator features when the discriminator is predicted to be retarded, such that the discriminator adaptively learns the temporally-vary distribution of generated data.

Our contributions are summarized as follows.

- From the perspective of online continual learning, we propose a novel approach that effectively improves the training of GANs by explicitly considering the time-varying distribution of generated samples.

- The proposed discriminator fast learns and adapts to the time-varying distribution of the generated samples via dynamically masking discriminator features, improving the training of GANs.

- Extensive experimental results show that our method generates high-quality images, outperforming state-of-the-art GAN approaches. Besides, the performance of our method surpasses advanced diffusion models.

## 2  Related Works

**Generative adversarial networks.**  GANs have attracted increasing attention from researchers [19, 84, 20, 5, 72]. By learning a min-max objective, GANs generate samples from a given data distribution, which can serve for various applications, *e.g.*, image editing [18, 28, 75, 3, 27, 97, 83, 43] and text-conditional image synthesis [69, 31, 78, 45, 64, 98]. As a fundamental technology, GANs coupled with the diffusion model (DM) [65, 26, 58, 62, 66] and auto-regressive model (AR) [63, 91, 10, 9] indirectly turn on the light of AIGC [8]. Compared with the other emergent generative models, GANs have the advantages of rapid inference [31], generating fixed objects [31, 65] and controllable latent distribution [35, 36, 34, 4, 23, 75]. Recently, as an *answer ball* against DM and AR, GigaGAN [31], StyleGAN-T[69] and GALIP [78] adapted GANs to large image-text datasets, demonstrating that GANs can be another viable solution for zero-shot text-to-image generation. However, as *"No Pain, No Gain"*, the payments of GANs are also high, *i.e.*, GANs suffer from unstable training and mode collapse. As a directed result, GANs are difficult to be equipped with some standard modules, such as dropout [76] and masking operator [10, 24], which strengthen the model generalization via introducing randomness into the training process, but further decrease the training stability. Existing studies [92, 74, 13, 96] illustrated that GANs learn from the stream, where the training distribution varies in the different time steps. Without any regularization, GANs may harmfully learn a bias on historical data and neglect the current data. Our goal in this paper is to downgrade the potential payments of GANs and integrate the masking operator into GANs to overcome overfitting on historical data.

**Regularization on discriminators.**  As a model helping the generator to align with real distribution, the discriminator plays a pivotal role to train GANs. Various methods have been proposed to improve GANs by redesigning discriminators [2, 21, 50, 93, 87, 30, 48, 96, 32, 57]. WGAN [2] employs the discriminator to measure the synthetic and natural samples with Wasserstein distance. Since WGAN requires Lipschitz continuity to ensure model convergence, weight clipping [2] and gradient penalty [21] were proposed to enhance the discriminators. Such technologies are then inherited by the family of StyleGAN [35, 36, 34]. Thanks to the advanced architecture [6, 36, 32], several regularization technologies, including ADA [33], APA [30], LC-Reg [82], DiffAugment [95], Con-Reg [93], AdaptiveMix [50], DAG [80], and MEE [48] were proposed to train GANs with low-data regime [50, 30, 33, 82, 80, 95] or combat mode collapse [48]. In addition to exploring training data, some studies redesigned the discriminator via proposing extra tasks [11, 79, 29, 88] and implementing large-scale pre-trained models [68, 41]. In this paper, the proposed method does not require additional training data, which can work as a plug-and-play module for existing discriminators. Compared with the previous regularization focusing on data augmentation, the proposed method investigates the capacity of the discriminator from a new perspective, *i.e.*, 'augmenting' the architecture to quickly adapt the current (new) knowledge. Note that the most related study to this paper is DynamicD [87], which dynamically adapts the capacity of the discriminator to the training data. However, DynamicD suffers from divergent solutions on different data regimes. In contrast, our method can help to re-adjust the attentions of discriminators and obtain consistent improvement against varying regimes.

## 3  Pilot Study

We first conduct a study to investigate the distribution shifts of generated samples over time during training. We then demonstrate the challenges posed by such a time-varying distribution to the fixed discriminator. We adopt a representative GAN, *i.e.*, StyleGAN-V2 [36], for the study, due to its advanced network architectures, training strategies, and impressive results.

**Analysis on generated distribution over time.** We train a StyleGAN-V2 on a widely-used dataset FFHQ [35], to study the distribution changes of its generated samples. In particular, we collect 5k generated samples at each time interval during training and then analyze the distribution of generated samples per time interval. The distribution is estimated with Fréchet Inception Distance (FID) [25]. More specifically, FID measures the distribution similarity between real and fake samples by

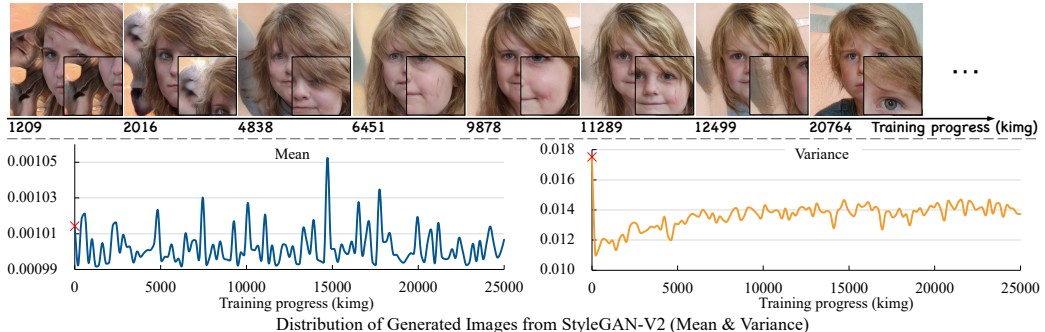

Figure 2: **Illustration of time-varying distributions of generated samples**, where we trace the training process of StyleGAN-V2 [36] on FFHQ [35]. The mean and variance of generated 5k samples' features are computed per time interval, showing the generated distributions are dynamic and time-varying during training, as the generator evolves. kimg refers to the number of images (measured in thousand) trained so far.

calculating the mean and covariance of real/fake samples. Similarly, we first extract the feature of a generated sample by an Inception model [77], and then represent the distribution of generated samples per time interval by the mean and variance of these samples' features.

We observe that the generated distributions undergo dynamical and complex changes over time (see Fig. 2), as the generator evolves during training. For example, the generated samples are not even independently and identically distributed (i.i.d) across the training progress in Fig. 2. Such time-varying distribution inevitably poses significant challenges to the discriminator, since this has not been explicitly considered in the discriminator typically.

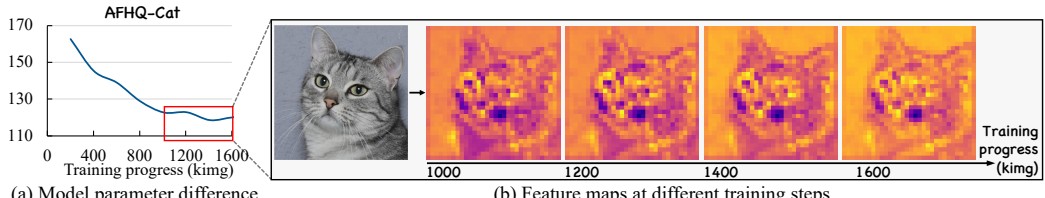

(a) Model parameter difference        (b) Feature maps at different training steps

Figure 3: We trace the training of StyleGAN-V2's discriminator. The curve in (a) shows the model parameter difference of the discriminator between $t_i$ and $t_{i-1}$ in the training progress. In (b), the attentive regions of the discriminator are almost fixed at training steps when the discriminator slows down its model parameter updating, where each feature map is represented by the feature space of the discriminator trained at a time step.

**Adaption ability of fixed discriminator.** Learning the distributions of generated samples is vital to training the generator. We investigate whether the fixed discriminator, which does not explicitly consider distribution changes, can effectively learn the time-varying distributions of generated samples. We argue that the discriminator needs to adaptively adjust its model weights and rely on different local features for discriminating generated samples at different steps. Suppose the generator synthesizes unrealistic hair for human faces at a time interval $[t_1, t_2]$, and synthesizes plausible hair but introduces artifacts in cheek regions at $[t_3, t_4]$ (see [1209, 4838] and [6451, 11289] kimgs in Fig. 2). It is expected that the discriminator pays less attention to hair regions and more attention to cheek regions in $[t_3, t_4]$, compared to $[t_1, t_2]$, so as to fast distinguish fake samples. Accordingly, the parameters of the discriminator model are expected to be updated, such that the discriminator adapts to the distribution changes of generated samples for $[t_1, t_2]$ to $[t_3, t_4]$.

Here we investigate the discriminator of StyleGAN-V2 since it is a representative fixed discriminator. We find the parameter differences of the discriminator are usually small between two adjacent steps, (see Fig. 3 (a)), indicating the discriminator slows down learning. Moreover, we find the discriminator model trained at different times pays attention to almost fixed local regions given an image, as indicated by the feature map extracted by the layers of the discriminator (see Fig. 3 (b)). As a result, the discriminator relies on old knowledge learned from historical data, which is insufficient to learn incoming generated samples under a dynamic distribution shift.

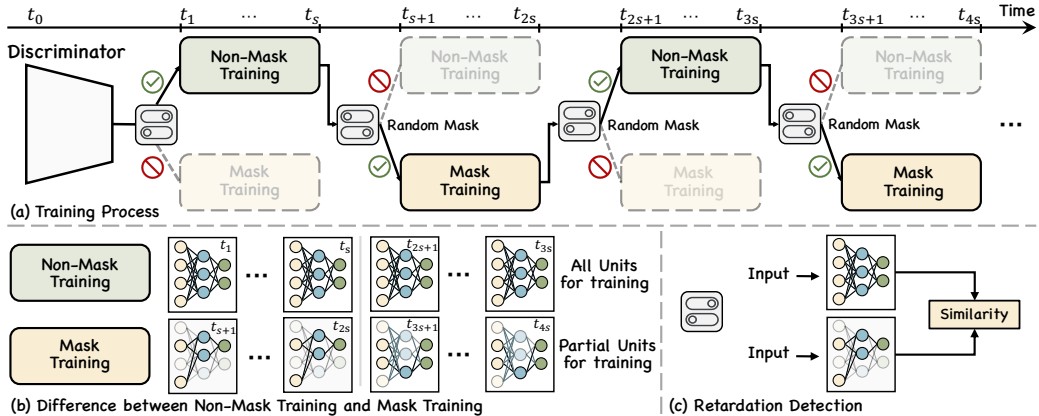

Figure 4: **The pipeline of the proposed method.** Our method, named Dynamically Masked Discriminator (DMD), automatically adjusts the discriminator via dynamically marking the discriminator. When DMD detects that the discriminator slows down learning, DMD dynamically assigns masks or removes masks to features of the discriminator per time interval, forcing the discriminator to learn new knowledge and preventing it from relying on old knowledge from historical samples.

## 4 Methodology

### 4.1 Problem Formulation

The discriminator learns the distribution of real and generated samples, which is vital to training the generator. Nevertheless, as studied in Sec. 3, the distribution of generated samples varies and drifts over time during training, posing challenges to existing discriminators (*e.g.*, [36, 50]). By treating the generated samples of the generator across training as a stream, we innovatively formulate learning the distribution of generated samples as an online continual learning problem. Motivated by recent advances in online learning [17, 7, 47], our target is to enable a discriminator to quickly adapt to incoming generated data.

Since the generator is trained via a min-max game with the discriminator, the distribution of generated samples is significantly complex and unpredictable. Hence, it is non-trivial to accurately predict distribution changes. In other words, it is difficult for the discriminator to predict when it needs to be rapidly adjusted for the new distribution.

**Overview.** We address the problem by considering two questions: (1) when does the discriminator slow down the learning from incoming generated samples? (2) how to force the discriminator to fast learn new knowledge? By exploring the two aspects, we propose a method named Dynamically Masked Discriminator (DMD) for GANs, as shown in Fig. 4.

### 4.2 Dynamically Masked Discriminator

Different from existing GANs methods [36, 50, 33, 48], we propose to automatically adjust the discriminator during training towards the time-varying distribution of generated samples. Instead of designing complex network architectures, we propose two key modules which can be easily integrated into existing discriminators. The two modules are (1) discriminator retardation detection and (2) dynamic discriminator adjustment. The first module automatically determines when the discriminator slows down learning. Here, *slow down* is referred to as the discriminator largely relies on old knowledge learned from historical data, given incoming generated samples with different distributions. The second module adjusts the discriminator for fast learning via dynamically assigning or removing masks at a certain interval.

With the two proposed modules, we improve an existing discriminator into a dynamic one at the time axis. That is, the discriminator in our method has two states, *i.e.*, mask and non-mask training. $\mathcal{D}$ denotes the original discriminator without masks, and $\bar{D}$ denotes the adjusted discriminator with masks. During training, we dynamically switch $\mathcal{D}$ and $\bar{D}$ for guiding the training of the generator

according to discriminator retardation detection. Let $\pi(\mathcal{D}|t)$ describe the probability to use $\mathcal{D}$ at time $t$. The probability of $\bar{D}$ is thus formed as $1 - \pi(\mathcal{D}|t)$. Let $\{\tilde{I}^t\}$ denote a stream consisting of samples generated by the generator $\mathcal{G}$, where $\tilde{I}^t = \mathcal{G}(z, \theta^t)$ is a generated sample from a vector $z$ at time $t$ during training, and $\theta^t$ denotes parameters of the generator at $t$. We formulate our training of GANs as follows:

$$\mathcal{L}_{\theta^t} := -\mathbb{E}_{z \sim p_z, t} \left[ \pi(\mathcal{D}|t) log(\mathcal{D}(\mathcal{G}(z, \theta^t), \phi^t)) + (1 - \pi(\mathcal{D}|t)) log(\mathcal{D}_M(\mathcal{G}(z, \theta^t), \phi^t)) \right] \quad (1)$$

$$\begin{aligned} \mathcal{L}_{\phi^t} := &-\mathbb{E}_{I \sim p_I, t} \left[ (1 - \pi(\mathcal{D}|t)) \, log(\mathcal{D}_M(I, \phi^t)) + \pi(\mathcal{D}|t) log(\mathcal{D}(I, \phi^t)) \right] \\ &- \mathbb{E}_{z \sim p_z, t} \left[ (1 - \pi(\mathcal{D}|t)) log(1 - \mathcal{D}_M(\mathcal{G}(z, \theta^t), \phi^t)) + \pi(\mathcal{D}|t) log(1 - \mathcal{D}(\mathcal{G}(z, \theta^t), \phi^t)) \right] \end{aligned} \quad (2)$$

where $I$ is a real sample and $\phi^t$ is the parameter of the discriminator at time $t$.

**Dynamic discriminator adjustment.** When the discriminator is detected to be retarded (*i.e.*, slow down learning), we adjust the discriminator to force it to learn fast. In this paper, we mainly focus on the issue that the discriminator largely relies on old knowledge learned from historical data and cannot rapidly adapt to incoming generated samples. To address this issue, one possible solution is to design complex network architectures. Instead, we propose to dynamically switch the discriminator from mask/non-mask states to non-mask/mask at a time interval. We argue that such dynamic masking at time intervals can break the original dependency of the discriminator on some local features that are important to distinguish historical samples, inspired by [15, 24, 46, 49, 76]. For example, by masking a feature map that is fed to a layer of the discriminator, we can control the layer of the discriminator and enforce it to pay attention to other regions of a generated image.

In this paper, we force the non-mask discriminator (*i.e.*, original one) $\mathcal{D}$ with parameter $\phi^{t_i}$ to be converted into the mask state $\mathcal{D}_M$ by masking its feature map or input. In particular, given the $d$-th layer of a discriminator, we propose to mask its input $\mathbf{F}^{(d-1),t}$ in a time interval $(t_j, t_{j+\epsilon}]$. Let $\mathbf{m}_{d-1}^t$ denote a mask in the $t$-th training step with the same size as $\mathbf{F}^{(d-1),t}$. We can dynamically mask the discriminator by masking the input features of its $d$-th layer:

$$\bar{\mathbf{F}}^{(d),t} = L(\mathbf{F}^{(d-1),t}, \mathbf{m}_{d-1}^t) = L(\mathbf{F}^{(d-1),t} \odot \mathbf{m}_{d-1}^t), \quad t \in (t_j, t_{j+\epsilon}] \quad (3)$$

where $\bar{\mathbf{F}}^{(d),t}$ is the output of $d$-th layer of a discriminator by masking, $\odot$ denotes Hadamard product, $L$ is the convolutional operator. Different from the traditional dropout, the mask does not continuously change. Instead, this mask is fixed for a period of training steps, *i.e.*, $(t_i, t_{i+\epsilon}]$. Switching the discriminator from mask to non-mask is simply removing all masks in the discriminator. Such dynamical switching encourages the discriminator to pay attention to various local regions/features over time, making our discriminator better adapt to the time-varying generated distribution and leading to better guidance of training the generator, compared with StyleGAN-V2 (see Fig. 1).

**Discriminator retardation detection.** This module is to detect whether the discriminator slows down the learning, *i.e.*, , the discriminator largely relies on old knowledge from historical data to distinguish future generated samples with new distributions. Given the current time step $t_i$, an ideal solution is to use future-generated samples with different distributions at time $(t_j, t_{j+\epsilon}]$ to evaluate the discriminator. However, these future-generated samples are unavailable at the current time step. Moreover, it is non-trivial to predict the distribution of future-generated samples.

Instead, we propose to construct a pseudo sample that possibly belongs to new distribution to detect the Retardation of the discriminator. Without loss of generality, we assume the distribution changes of generated samples correspond to the changes in local regions/features of these samples. For example, some hair regions are unrealistic in a generated sample at time $t_i$, and become realistic at $t_j$, as the generator evolves in Fig. 2. By randomly masking a generated sample or its intermediate features in the discriminator, it is possible to remove discriminative local regions/features that are important for the sample's distribution, shirting the sample to a new distribution. For example, if the right face region of the man is removed/masked in Fig. 5, the remaining regions become more realistic (*i.e.*, belongs to a new distribution).

Therefore, given a generated sample $\tilde{I}^{t_i}$ for a distribution, we construct a pseudo sample that possibly belongs to another distribution by randomly masking local image regions or intermediate features of the discriminator. For generated samples at time $t$, if the discriminator determines they are

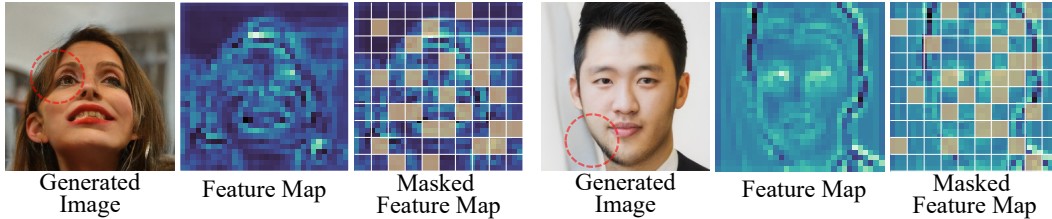

| Generated Image | Feature Map | Masked Feature Map | Generated Image | Feature Map | Masked Feature Map |

Figure 5: **Illustration of the generated images, feature maps, and the corresponding masks.** The feature maps are extracted from the discriminator of StyleGAN-V2 [36] trained on FFHQ [35], and the red region denotes the artifacts. By masking feature maps of the discrimination, it is possible to remove discriminative local regions/features (*e.g.*, unrealistic regions) that are important for the sample's distribution, shirting the sample to a new distribution.

similar to their pseudo samples, it is probable that the discriminator does not discriminate the distribution difference and is detected as retardation. Formally, given $m$ samples at time $t$, we can define our heuristic Retardation metric, $\mathcal{R}_t$, as

$$\mathcal{R}_t = \frac{1}{m} \sum_{i \in U_t} \frac{\bar{\mathbf{F}}_i^{(d),t} \cdot \mathbf{F}_i^{(d),t}}{|\bar{\mathbf{F}}_i^{(d),t}||\mathbf{F}_i^{(d),t}|} \qquad (4)$$

where $U_t$ is the index of samples, $\bar{F}_i^{(d),t}$ is the output of the $d$-th layer of the discriminator taking original sample as input, $\bar{F}_i^{(d),t}$ is that of a pseudo sample where its image region or feature maps in the discriminator is masked. If $\mathcal{R}_t$ is larger than a predefined threshold, the discriminator doesn't detect differences between original and pseudo samples and is detected as retardation. The pseudo-code is given in the *Appendix*.

## 5 Experiments

### 5.1 Experimental Setup

**Dataset.** We evaluate the performance of our method on six widely used datasets, including AFHQ-Cat [12], AFHQ-Dog [12], AFHQ-Wild [12], FFHQ [35], and LSUN-Church [90] with 256 × 256 resolutions and CIFAR-10 [39] with 32 × 32 resolutions. We elaborate on these datasets and implementation details in the *Appendix*.

**Evaluation Metrics.** Following prior works [50, 33, 30], we use two evaluation metrics, including Fréchet Inception Distance (FID) [25] and Inception Score (IS) [67], to evaluate the quality of generated results. The number of testing samples is set to 50k in our experiments.

**Baselines.** Following the previous studies [33, 30, 50], we integrate our method with StyleGAN-V2 [36], StyleGAN-V3 [34], and BigGAN [6]. We compare our method with state-of-the-art methods that improve discriminators via data augmentation, including ADA [33] and APA [30]. We also compare with GANs using regularization: LC-Reg [82], zCR [93], InsGen [88], Adaptive Dropout [33], AdaptiveMix [50], MEE [48] and DynamicD [87]. Besides StyleGAN, we compare with other generative models: DDPM [26], ImageBART [16], PGGAN [32] and LDM [65].

### 5.2 Comparison with State-of-the-art Methods

**Main results.** To show the superiority of the proposed method, we compare the performance of our approach with state-of-the-art methods on FFHQ. As shown in Table 1, our method achieves the best performance in terms of FID score on FFHQ. Recent methods [87, 50, 88] improve GANs by data augmentation, *e.g.*, ADA [33] and APA [30]. By further combining with APA [30], FID score of our method reduces from 3.299 to 3.075, significantly surpassing the other approaches. In addition to the quantitative analysis, Fig. 6 and Fig. 7 show qualitative results of our method. StyleGAN-V2 introduces noticeable artifacts, while our method generates images with much better quality. This is because our discriminator effectively learns the time-varying distributions of generated samples,

Table 1: Our method compared to the generative models and the variants of StyleGAN-V2 [36] on FFHQ [35]. Comparisons are taken from [50, 87, 65].

| Methods | FID↓ |
|---|---|
| ImageBART [16] (NIPS'21) | 9.57 |
| U-Net GAN (+aug) [73] (CVPR'20) | 10.9(7.6) |
| LDM [65] (CVPR'22) | 4.98 |
| StyleGAN-V2 [36] (CVPR'20) | 3.862 |
| StyleGAN-V2 (Re-Impl.) | 3.810 |
| DiffAugment [95] (NIPS'20) | 4.840 |
| Adaptive Dropout [33] (NIPS'20) | 4.160 |
| LC-Reg [82] (CVPR'21) | 3.933 |
| DynamicD-Decreasing [87] (NIPS'22) | 3.740 |
| ADA [33] (NIPS'20) | 3.880 |
| ADA (Re-Impl.) [33] (NIPS'20) | 3.713 |
| APA [30] (NIPS'21) | 3.678 |
| MEE (Re-Impl.) [48] (AAAI'23) | 3.626 |
| AdaptiveMix [50] (CVPR'23) | 3.623 |
| AdaptiveMix (+APA) [50] (CVPR'23) | 3.609 |
| DynamicD-Increasing [87] (NIPS'22) | 3.530 |
| zCR [93] (ICLR'20) | 3.450 |
| InsGen (+ADA) [88] (NIPS'21) | 3.310 |
| Ours: DMD | 3.299 |
| Ours: DMD (+APA) | **3.075** |

Table 2: FID ↓ of our method compared with other regularizations for GAN training on AFHQ-V2 [12]. Comparison results are taken from [87, 50].

| Methods | Cat | Dog | Wild |
|---|---|---|---|
| StyleGAN-V2 [36] (CVPR'20) | 7.924 | 26.310 | 3.957 |
| LC-Reg [82] (CVPR'21) | 6.699 | - | - |
| ADA [33] (NIPS'20) | 6.360 | 18.930 | 3.800 |
| DynamicD (+ADA) [87] (NIPS'22) | 5.410 | 16.000 | 3.340 |
| AdaptiveMix [50] (CVPR'23) | 4.477 | - | - |
| MEE (Re-Impl.) [48] (AAAI'23) | **4.453** | - | - |
| Ours: DMD (+ADA) | 5.223 | **12.764** | **3.261** |
| StyleGAN-V2 (+APA) | 4.645 | 13.913 | 2.919 |
| Ours: DMD (+APA) | **4.261** | **11.784** | **2.625** |

Table 3: Our method compared to other state-of-the-art generative models on LSUN-Church [90]. We re-implement StyleGAN-V2, and the other results are from LDM [65].

| Methods | FID↓ |
|---|---|
| DDPM [26] (NIPS'20) | 7.89 |
| ImageBART [16] (NIPS'21) | 7.32 |
| PGGAN [32] (ICLR'18) | 6.42 |
| StyleGAN-V2 (Re-Impl.) [36] (CVPR'20) | 4.29 |
| LDM [65] (CVPR'22) | 4.02 |
| DynamicD [87] (NIPS'22) | 3.87 |
| Ours: DMD | **3.06** |

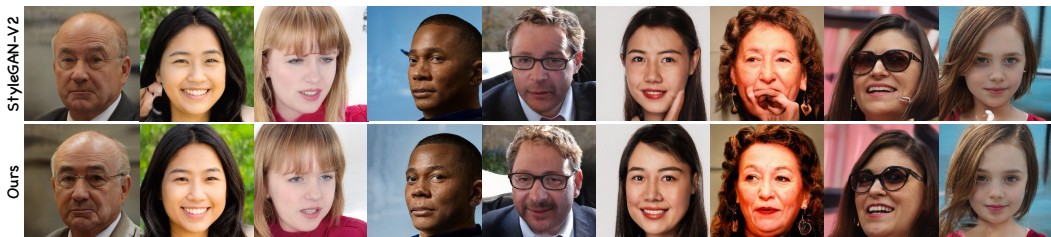

Figure 6: Qualitative comparison results of our method and StyleGAN-V2 on FFHQ dataset.

which improves the training of the generator. (More experimental results and details are provided in the *Appendix*.)

We also perform comparison experiments on dataset AFHQ-V2. As listed in Table 2, our method outperforms state-of-the-art approaches, boosting the performance on all three sub-datasets, showing the effectiveness of our method in improving the training of the generator. DynamicD [87] empirically increases/decreases the capacity of the discriminator for GAN training, which is most related to our work. However, our method outperforms DynamicD by a large margin. For example, FID of DynamicD is 16.000 on the dog dataset, and our method achieves better FID (11.784). We also evaluate our method on StyleGAN-V3 [34] and BigGAN [6] respectively in Table 4 and Table 5, which show consistent improvements by adding our proposed method into these two baselines.

Table 4: Our method over StyleGAN-V3 [34] on AFHQ-Cat [12].

| | FID ↓ | IS ↑ |
|---|---|---|
| StyleGAN-V3R | 7.616 | 1.881 |
| StyleGAN-V3R w/ Ours(DMD) | **6.864** | **1.915** |
| StyleGAN-V3T | 5.850 | 1.916 |
| StyleGAN-V3T w/ Ours(DMD) | **4.921** | **1.969** |

Table 5: Our method over BigGAN [6] on CIFAR-10 [39].

| | FID ↓ |
|---|---|
| Unconditional BigGAN | 16.040 |
| Unconditional BigGAN w/ MEE [48] | 13.860 |
| Unconditional BigGAN w/ Ours(DMD) | **10.540** |

**Performance on the large-scale training dataset.** We evaluate the performance of our method trained on a large-scale dataset LSUN-Church [90] shown in Table 3. LSUN-Church contains 120k images and can be augmented to 240k with flipping. Diffusion models and auto-regressive models achieve impressive performance by training on large-scale datasets. We compare our method with representative diffusion models: LDM [65], DDPM [26], and ImageBART [16]. LDM outperforms StyleGAN-V2 slightly (4.02 vs. 4.29 FID). However, our method significantly reduces the FID of StyleGAN-V2 from 4.29 to 3.06 FID, achieving the best performance, compared with the other generative models.

## 5.3 Ablation Study and Analysis

**Ablation studies.** Since our method is plug-and-play, which is easy to be integrated into existing methods. We adopt StyleGAN-V2 as the baseline and evaluate the improvement of integration with our method. As shown in Table 6, our method improves StyleGAN-V2 on AFHQ-V2 and LSUN datasets. For example, our method enables better performance of StyleGAN-V2, where FID is improved by 25.8% on AFHQ-Cat. One of our contributions is to automatically switch the discriminator between mask and non-mask training, according to discriminator retardation detection. We evaluate the effectiveness of this automatic scheme. We test the performance of the baseline with different fixed intervals for the transition between the mask and non-mask training. As shown in Table 7, fixed interval improves the performance of StyleGAN-V2, indicating that dynamically adding/removing masks in the discriminator helps the training of GANs. Nevertheless, the proposed method can achieve the best performance, since discriminator retardation detection automatically detects when the discriminator slows down.

Table 6: The ablation study of our method on AFHQ-V2 [12] and LSUN-Church [90], where the baseline is the StyleGAN-V2 [36] and Ours is StyleGAN-V2+DMD.

|  | AFHQ-Cat | | AFHQ-Dog | | AFHQ-Wild | | LSUN-Church | |
|---|---|---|---|---|---|---|---|---|
|  | FID ↓ | IS ↑ | FID ↓ | IS ↑ | FID ↓ | IS ↑ | FID ↓ | IS ↑ |
| Baseline | 7.924 | 1.890 | 26.310 | 9.000 | 3.957 | 5.567 | 4.292 | 2.589 |
| Ours | 5.879(-25.8%) | 1.988(+5.2%) | 21.240(-19.3%) | 9.698(+7.8%) | 3.471(-12.3%) | 5.647(+1.4%) | 3.061(-28.7%) | 2.792 (+7.8%) |

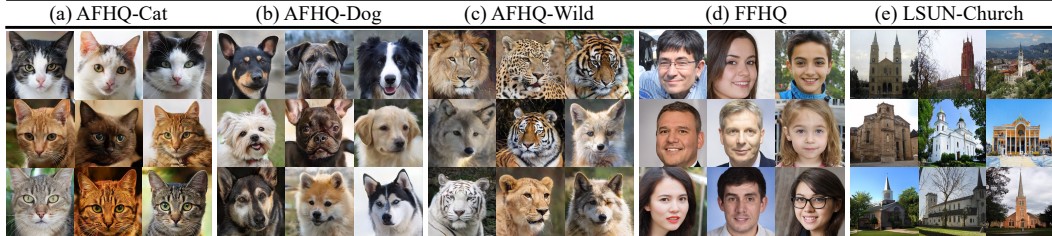

| (a) AFHQ-Cat | (b) AFHQ-Dog | (c) AFHQ-Wild | (d) FFHQ | (e) LSUN-Church |

Figure 7: The generated samples of the proposed method on (a) AFHQ-Cat, (b) AFHQ-Dog, (c) AFHQ-Wild, (d) FFHQ, and (e) LSUN-Church. All training data are in a resolution of $256 \times 256$.

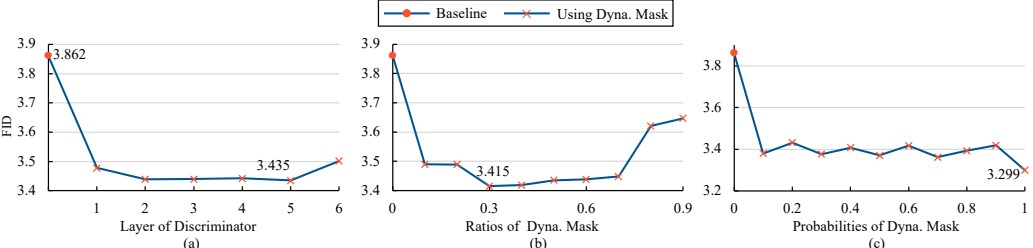

Figure 8: **Probe of hyper-parameters of the proposed method on FFHQ-70k.** We discuss the depth, ratio, and probabilities for masking. Probabilities of Dyna. Mask control the times in the masking stage. In each iteration of mask training, masking the discriminator in the 5-th layer with 0.3 ratios can achieve the best result.

**Hyper-parameters and masking strategies.** We then search the hyper-parameters for the proposed method. As shown in Fig. 8, we separately investigate the efficacy of the depth for masking, the mask ratio, and the probabilities to trigger the mask. Accordingly, the best performance appears with the 5-th layer, 0.3 masking ratios, and masking discriminator every time (*i.e.*, probability=1) in the mask training stage. Meanwhile, we investigate different masking strategies, including Vanilla Dropout, Input Masking, Dynamic Head, and Vanilla Dropout. We detail the corresponding implementations in the *Appendix*. As shown in Table 8, the proposed method can achieve the best performance among all the cases. Note that the empirical study shows that directly applying dropout will incur unstable training and hurt the capacity of the discriminator, leading to worse FID. This shows the advantage of our method, which automatically switches the discriminator at a time interval.

Table 7: FID ↓ of our method compared to the StyleGAN-V2 with fixed interval mask on FFHQ-70k.

| Inverval (kimg) | FID |
|---|---|
| 4 | 3.823 |
| 8 | 3.362 |
| 16 | 3.365 |
| 24 | 3.391 |
| DMD (Ours) | **3.299** |

Table 8: FID ↓ of and the proposed method on AFHQ-V2 [12] and other masking strategies for StyleGAN-V2[36]. Note that Vanilla Dropout [76] is based on a 0.5 drop rate.

| | Cat | Dog | Wild |
|---|---|---|---|
| StyleGAN-V2 [36] | 7.924 | 26.310 | 3.957 |
| Attention-based Mask | 7.294 | 23.782 | 3.800 |
| Input Masking | 6.809 | 22.054 | 3.515 |
| Dynamic Head | 7.388 | 23.050 | 3.736 |
| Vanilla Dropout [76] | 9.294 | 28.099 | 5.056 |
| DMD (Ours) | **5.879** | **21.240** | **3.471** |

## 6 Conclusion

In this paper, we propose a novel method named DMD for training GANs from a new perspective, *i.e.*, online continuous learning. Our study shows that the distribution of generated samples is time-varying, while this problem has been underexplored. This makes the discriminator often largely rely on historically generated data, instead of learning new knowledge from the incoming generated samples, degrading generation performance. We propose to force the discriminator to fast learn and adapt to incoming generated samples. We propose a simple yet effective method that automatically detects whether the discriminator slows down learning, and adjusts the discriminator by dynamically imposing or removing masks of the discriminator per time interval. Experimental results show our method improves the learning of the discriminator on temporally-vary distribution, boosting the guidance of training the generator, achieving the best performance than state-of-the-art methods.

**Limitations:** Theoretical studies can make this work more comprehensive; however, we have not explored it in the paper, since it is beyond the scope of this study. Moreover, while the proposed method can effectively improve the training of the CNN-based GANs models, combining our method with Transformer-based ones is left to be investigated in the future.

**Broader Impact:** Our method can be used for various applications such as producing training data and creating photorealistic images. On the other hand, like other generative models, our method can be misused for the application of Deepfake [1], where fake content is synthesized to deceive and mislead people, leading to a negative social impact. Nevertheless, many researchers have considered this problem while exploring fake content detection and media forensics techniques. In addition, we believe there would be regulations on fake content generation, such as forcing synthesized content to be injected with identifications that indicate it to be fake.

## Acknowledgments and Disclosure of Funding

We thank Hasan Abed Al Kader Hammoud for his valuable constructive suggestions and help. This work was supported by the SDAIA-KAUST Center of Excellence in Data Science and Artificial Intelligence (SDAIA-KAUST AI).

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

# Appendix

## A  Introduction

In this appendix, we first introduce the implementation details of our Dynamic Mask Discriminator (DMD), and then provide more technical details of the proposed DMD in Sec. B. In Sec. C.1 and C.2, we elaborate on the additional details of datasets and experiment settings. In Sec. C.3, we provide additional experimental results. We also report the error bars of our method in Sec. D.

## B  Implementation Details

In this section, we provide the implementation details of our proposed method. We have utilized the PyTorch [59] public platform to conduct our experiments. Our models were trained on a workstation equipped with 8 NVIDIA Tesla V100 GPUs with 32 GB memory, a CPU of 2.8GHz, and 512GB RAM.

### B.1  More details of the proposed method.

Our method is designed to automatically detect the retardation of the discriminator and force it to fast learn the new knowledge given time-varying distributions of generated samples. To show the detailed strategy of the proposed DMD method, the pseudo-code is given in Algorithm. 1. In our main paper, we have demonstrated the effectiveness of our proposed DMD method by integrating it with image generation techniques on StyleGAN-V2 [36].

Since the retardation of learning new knowledge can potentially be addressed by augmenting the input stream or the outcome of the discriminator, we also extend another two schemes to achieve dynamic discriminator adjustment. Besides the proposed dynamic feature masking scheme, we can dynamically mask the input and output of the discriminator when it is detected as retardation. The corresponding details are shown summarized in Algorithm. 2 and Algorithm. 3.

### B.2  More details of model parameter difference

We detail *model parameter difference* which is used in the main paper Figure 3(a) for investigating the fixed discriminator of StyleGAN-V2. Model parameter difference metric is to measure the differences of the discriminator model weights between two adjacent training steps:

$$\mathbf{d}_{t_j} = \|W_{t_j}^d - W_{t_{j-1}}^d\|^2 \tag{5}$$

where $W_{t_j}^d$ is the parameter weight of $d$-th layer of the discriminator during $t_j$ training step, and $\|\cdot\|^2$ is l2 norm. A smaller $\mathbf{d}_{t_j}$ indicates that the parameter weights of $d$-th layer at $t_j$ training step are more similar to that at $t_{j-1}$ training step, *i.e.*, parameter weights are updated slowly. In other words, given new arrival generated samples with new distributions, a smaller $\mathbf{d}_{t_j}$ indicates that the discriminator slows down the learning of new knowledge to some extent. In the main paper, we calculate $\mathbf{d}_{t_j}$ in the full connection layer of the discriminator.

## C  Additional Details on Experiments

### C.1  Datasets and Experimental Settings

**AFHQ-V2 [12]** consists of 3 independent sub-datasets, which include around 5,000 closeups of cat, dog, and wildlife faces, respectively (denoted as **AFHQ-Cat**, **AFHQ-Dog**, and **AFHQ-Wild**). We utilized a high-quality Lanczos filter [42] to resize all images to a resolution of $256 \times 256$. We then conducted experiments on three sub-datasets while setting StyleGAN-V2 [36] as the baseline model. We maintain consistency with ADA [33], by using identical network architectures [36], weight demodulation [36], style mixing regularization [35], path length regularization, lazy regularization [36], equalized learning rate for all trainable parameters [32], non-saturating logistic loss [19] with $R_1$ regularization [54], and the Adam optimizer [37].

---

**Algorithm 1** Dynamic Mask Discriminator

---

**Require:**
  Generator $\mathcal{G}_{\theta^t}$; Current Discriminator $D_\phi(\cdot)$; Non-Masked Discriminator $\mathcal{D}_\phi(\cdot)$; Dynamically Masked Discriminator $\bar{D}_\phi(\cdot)$; Training Step $t$;
  $d$-th Layer of Discriminator $\mathbf{F}^{(d),t}$; Dynamic Mask $\mathbf{m}_d^t$;
  $d$-th Masking Layer of Discriminator $\bar{\mathbf{F}}^{(d),t}$; Predefined Threshold $\lambda$; Retardation Metric $R_t$;
  Set $U_t$ Containing $m$ Samples for Calculating Retardation Metric;
  The number of training steps $n_t$; The number of images per training step $n_s$;

**Ensure:**
  Initialize $R_t \leftarrow 0$ and $t \leftarrow 1$; Random $\mathbf{m}_d^t$;

1: **while** $\theta$ has not converged **do**
2:    **for** $t = 1$ to $n_{\mathrm{t}}$ **do**
3:       **if** $R_t > \lambda$ **then**
4:          $\mathbf{M}^t \leftarrow \mathbf{m}_d^t$; $\mathbf{M}_T \leftarrow \mathbf{M}^t$; $D_\phi(\cdot) \leftarrow \bar{D}_\phi(\cdot)$
5:       **else**
6:          $\mathbf{M}^t \leftarrow \mathrm{vector}(1)$; $\mathbf{M}_T \leftarrow \mathbf{m}_d^{t+1}$; $D_\phi(\cdot) \leftarrow \mathcal{D}_\phi(\cdot)$
7:       **end if**
8:       **for** $s = 1$ to $n_s$ **do**
9:          $\bar{\mathbf{F}}^{(d),t} \leftarrow \mathbf{F}^{(d),t} \odot \mathbf{M}^t$
10:         $L_{\phi(t)} \leftarrow -\mathbb{E}_{I,t}[log(D_\phi(I))] - \mathbb{E}_{z \sim p_z,t}[log(1 - D_\phi(\mathcal{G}(z,\theta^t)))]$
11:         $\theta_s \leftarrow \mathrm{Adam}(\frac{\partial\phi(t)}{\partial\theta_{s-1}})$;
12:       **end for**
13:       $\mathcal{R}_t = \frac{1}{m}\sum_{i \in U_t}\frac{\bar{\mathbf{F}}_i^{(d),t} \cdot \mathbf{F}_i^{(d),t}}{|\bar{\mathbf{F}}_i^{(d),t}||\mathbf{F}_i^{(d),t}|}$
14:    **end for**
15: **end while**
16: Return $\theta$;

---

**FFHQ [35]** comprises 70,000 images of human faces, which we used for training after downscaling them to a resolution of $256 \times 256$. In this case, we set StyleGAN-V2 [36] as the baseline and used the same settings as those for AFHQ-V2.

**LSUN-Church [90]** includes 126,000 images of outdoor church. We downscale them to $256 \times 256$ as the training data. In this case, we also set StyleGAN-V2 [36] as the baseline and used the same settings as those for FFHQ.

## C.2  Baselines

In accordance with previous studies [33, 30, 50], we have integrated our proposed method with StyleGAN-V2 [36]. In order to assess the effectiveness of our method, we have compared it with state-of-the-art methods that improve discriminators through data augmentation, including ADA [33] and APA [30]. We have also compared our method with GANs that utilize regularization techniques, such as LC-Reg [82], zCR [93], InsGen [88], Adaptive Dropout [33], AdaptiveMix [50], MEE [48], and DynamicD [87].

## C.3  Additional Experimental Results

**Comparing our method with other masking strategies.** Besides the masking strategies discussed in the paper, another masking strategy, namely Continualy-Changed-mask-ratio Discriminator (CCD) is explored here, which replaces our dynamic discriminator adjustment module by gradually increasing the mask ratio from 0.1 to 0.9 or decreasing the mask ratio from 0.9 to 0.1 over time. We evaluate the CCD with StyleGAN-V2 on the AFHQ-Cat ($256 \times 256$ pixels) dataset. Compared with the StyleGAN-V2 (7.924 FID), CCD with StyleGAN-V2 achieves 8.441 FID. However, our method (5.879 FID) still outperforms CCD, since CCD increases instabilities in GAN training. More specifically, the process of GAN training is to play a min-max two-player game between the generator and discriminator, which is more unstable than typical classification problems. Compared with our method, CCD causes the discriminator's discrimination ability to be more

---

**Algorithm 2** Dynamic Mask Discriminator Assert in Input (***Input Masking***)

---

**Require:**
    Generator $\mathcal{G}_{\theta^t}$; Discriminator $\mathcal{D}_\phi(\cdot)$; Training Step $t$; Dynamic Mask $\mathbf{m}^t$;
    Predefined Threshold $\lambda$; Retardation Metric $R_t$;
    Set $U_t$ Containing $m$ Samples for Calculating Retardation Metric;
    $d$-th Layer of Discriminator $\mathbf{F}^{(d),t}$; After Masking Input $\bar{\mathbf{F}}^{(d),t}$ ;
    The number of training steps $n_t$; The number of images per training step $n_s$;
**Ensure:**
    Initialize $R_t \leftarrow 0$ and $t \leftarrow 1$; Random $\mathbf{m}^t$;
1:  **while** $\theta$ has not converged **do**
2:      **for** $t = 1$ to $n_{\mathrm{t}}$ **do**
3:         **if** $R_t > \lambda$ **then**
4:             $\mathbf{M}^t \leftarrow \mathbf{m}^t$; $\mathbf{M}_T \leftarrow \mathbf{M}^t$;
5:         **else**
6:             $\mathbf{M}^t \leftarrow \text{vector}(1)$; $\mathbf{M}_T \leftarrow \mathbf{m}^{t+1}$;
7:         **end if**
8:         **for** $s = 1$ to $n_s$ **do**
9:             $\bar{I} \leftarrow I \odot \mathbf{M}^t$; $\bar{\bar{I}} \leftarrow \mathcal{G}(z, \theta^t) \odot \mathbf{M}^t$
10:           $L_{\phi(t)} \leftarrow -\mathbb{E}_{\bar{I},t}[log(D(\bar{I}))] - \mathbb{E}_{z\sim p_z,t}[log(1 - D(\bar{\bar{I}}))]$
11:           $\theta_s \leftarrow \text{Adam}(\frac{\partial \phi(t)}{\partial \theta_{s-1}})$;
12:         **end for**
13:         $\mathcal{R}_t = \frac{1}{m}\sum_{i\in U_t} \frac{\bar{\mathbf{F}}_i^{(d),t} \cdot \mathbf{F}_i^{(d),t}}{|\bar{\mathbf{F}}_i^{(d),t}||\mathbf{F}_i^{(d),t}|}$
14:      **end for**
15: **end while**
16: Return $\theta$;

---

frequently changed and uncertain, making it more difficult for the generators to fool the discriminator. The deteriorated performance of the generator would further negatively affects the training of the discriminator. Instead, when the improvement of the discriminator slows down, our method switches from non-mask training to mask training, otherwise maintains masking/non-masking at a time interval, which provides more stable training than CCD.

**Experiments on AFHQ-Cat with 512×512 pixels.** We conduct experiments on the high-resolution AFHQ-Cat dataset, where the resolution of an image is 512×512 pixels. By integrating our method with StyleGAN-V2, our method reduces the FID of StyleGAN-V2 by a margin of 18.77% (from 4.3160 to 3.5061), effectively improving the performance of StyleGAN-V2 on higher-resolution images.

**Experiments on ImageNet-1000.** We integrate the proposed DMD with an intermediate checkpoint of the BigGAN [6] model which has been trained for 100,000 iterations. We then further trained models from 100,000 to 107,500 iterations on ImageNet-1000 at 128×128 resolution using two NVIDIA V100 GPUs. Compared with the original BigGAN that achieves the FID of 12.213 at the 107,500 iteration, our method (i.e., DMD) improves the training of BigGAN (FID of 11.212), outperforming the original BigGAN by 8.196%. This not only shows that our method benefits the training of BigGAN on large-scale datasets, but also demonstrates the flexibility and compatibility of our method in combination with GAN models such as pre-trained GAN models.

**Experiments on historical knowledge retention of discriminator.** We conduct experimental studies inspired by the study in online continual learning for rapid adaptation [7]. Given the discriminator trained from the beginning to time $T$, we evaluate its historical knowledge retention by evaluating its discrimination performance on historical data generated at time $T - t$, and evaluate its rapid adaptation by evaluating its performance in current data at time $T$ and future data at $T + t$.

Table 9 shows that the discriminator of StyleGAN-V2 achieves high accuracy on historical data, however, performs much worse on current and future data. This shows that the discriminator of StyleGAN-V2 retains historical knowledge (i.e., high accuracy in historical data), yet, does not fast learn and adapt to the changes in the distribution of future data.

**Algorithm 3** Dynamic Mask Discriminator Assert in Outcome Logits (***Dynamic Head***)

---

**Require:**
    Generator $\mathcal{G}_{\theta^t}$; Discriminator $\mathcal{D}_\phi(\cdot)$; Training Step $t$;
    Outcome Logit Number of $\mathcal{D}_\phi(\cdot)$ and $\bar{\mathcal{D}}_\phi(\cdot)$ $k$; Dynamic Mask $\mathbf{m}^t$;
    $d$-th Layer of Discriminator $\mathbf{F}^{(d),t}$; After Masking Outcome Logits $\bar{\mathbf{F}}^{(d),t}$;
    Predefined Threshold $\lambda$; Retardation Metric $R_t$;
    Set $U_t$ Containing $m$ Samples for Calculating Retardation Metric;
    The number of training steps $n_t$; The number of images per training step $n_s$;
**Ensure:**
    Initialize $R_t \leftarrow 0$ and $t \leftarrow 1$; Random $\mathbf{m}^t$;
1: **while** $\theta$ has not converged **do**
2:     **for** $t = 1$ to $n_{\text{t}}$ **do**
3:         **if** $R_t > \lambda$ **then**
4:             $\mathbf{M}^t \leftarrow \mathbf{m}^t$; $\mathbf{M}_T \leftarrow \mathbf{M}^t$;
5:         **else**
6:             $\mathbf{M}^t \leftarrow \text{vector}(1)$; $\mathbf{M}_T \leftarrow \mathbf{m}^{t+1}$;
7:         **end if**
8:         **for** $s = 1$ to $n_s$ **do**
9:             $L_{\phi(t)} \leftarrow -\mathbb{E}_{I,t}[log(\sum(D(I) \odot \mathbf{M}^t)))] - \mathbb{E}_{z \sim p_z, t}[log(1 - \sum(D(\mathcal{G}(z, \theta^t)) \odot \mathbf{M}^t))$
10:             $\theta_s \leftarrow \text{Adam}(\frac{\partial \phi(t)}{\partial \theta_{s-1}})$;
11:         **end for**
12:         $\mathcal{R}_t = \frac{1}{m} \sum_{i \in U_t} \frac{\bar{\mathbf{F}}_i^{(d),t} \cdot \mathbf{F}_i^{(d),t}}{|\bar{\mathbf{F}}_i^{(d),t}||\mathbf{F}_i^{(d),t}|}$
13:     **end for**
14: **end while**
15: Return $\theta$;

---

Instead, by detecting retardation and masking the discriminator, our method enforces the discriminator to reduce retained historical knowledge (see decreased accuracy on historical data), while effectively encouraging it to fast learn and adapt to new knowledge of future data (i.e., our method achieves higher accuracy on current and future data than StyleGAN-V2).

Table 9: The accuracy of the discriminator on historical data and future data, where the discriminator is trained at 800 kimgs step, and $(\cdot)$ indicates the data at time $T - t$ or $T + t$.

| Method | Historical data (200) | Historical data (600) | Current data (800) | Future data (1000) |
|---|---|---|---|---|
| StyleGAN-V2 | 0.9696 | 0.9488 | 0.8236 | 0.604 |
| Ours (StyleGAN-V2 + DMD) | 0.9129 | 0.9229 | 0.8793 | 0.6543 |

In addition, we calculate the average gradient of the StyleGAN-V2 during the training phase in Table 10, showing the gradient does not vanish from 0 to 1200 kimgs.

Table 10: Average gradient of the StyleGAN-V2 during training.

| kimgs | 0 | 200 | 400 | 600 | 800 | 1000 | 1200 |
|---|---|---|---|---|---|---|---|
| Gradient | 3.5145e-08 | 4.6110e-07 | -9.4235e-07 | 5.6989e-07 | -4.3328e-07 | 9.2649e-08 | 1.2138e-06 |

**Model parameter difference of the proposed method.** We also compute the model parameter difference of the proposed method in the training of the AFHQ-Cat dataset. Our method detects the discriminator to be retarded at 1000 kimgs after 800 kimgs. Our method then applies the proposed DMD adjustment to train StyleGAN-V2 from 1000 to 1004 kimgs, which increases the model difference of our method by 63.889%, compared with the model difference between 800 to 1000 kimgs. Instead, the model difference of StyleGAN-V2's discriminator is decreased by 2.778% without our method. These results show StyleGAN-V2's discriminator slows down learning and our method effectively encourages the discriminator to fast learn.

**Computation cost.** Our method incurs a negligible increase in memory cost, since our method only additionally detects discriminator retardation and introduces feature masks. For computational cost, our method does not affect the inference time, since our method is designed for the discriminator and only the generator is used during inference. For training, our method introduces moderate time cost; however, this is affordable (e.g., increasing GPU number to reduce training time), and the additional training time cost depends on the time interval of estimating the retardation. Our method additionally introduces 6 hours 11 minutes for training on AFHQ in our experiments, when our method is integrated with StyleGAN-V2 and retardation is detected at every 4 kimgs. When detecting retardation at every 20 kimgs, our method only additionally takes 1.3 hours for training.

**Additional Generated Distribution.** Our main paper studies the generated distributions of StyleGAN-V2. Here, we additionally provide the distribution of the generated samples of APA [30] method on FFHQ[35] in Fig. 9. Fig. 9 also indicates that the generated distributions undergo dynamical and complex changes over time as the generator evolves during training. As a result, the generated samples are not independently and identically distributed (i.i.d) across the training progress, posing significant challenges in learning the generated distributions.

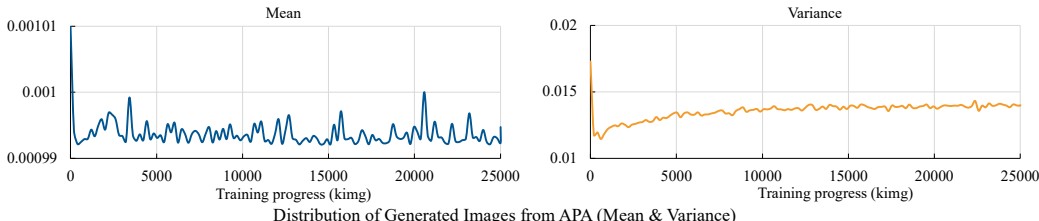

Distribution of Generated Images from APA (Mean & Variance)

Figure 9: Illustration of time-varying distributions of generated samples in the training process of APA [30] on FFHQ [35]. The mean and variance generated samples' features show the generated distributions are dynamic and time-varying during training, as the generator evolves.

# D  Error Bar

To evaluate the reproducibility of our method's results, we run our experiments three times using random seeds and the same hyper-parameters. Table 11 and Table 12 list the mean and variance of experimental results to show the error bar.

As shown in Table 11 and Table 12, our method performs stably on multiple datasets *i.e.*, AFHQ-V2, FFHQ and LSUN-Church datasets, indicating the reproducibility of our method.

Table 11: Quantitative results of our method on AFHQ-V2 dataset [12], error bars are reported in terms of mean and variance, and Ours is StyleGAN-V2+DMD.

|  | AFHQ-Cat | | AFHQ-Dog | | AFHQ-Wild | |
|---|---|---|---|---|---|---|
|  | FID ↓ | IS ↑ | FID ↓ | IS ↑ | FID ↓ | IS ↑ |
| StyleGAN-V2 | 7.924 | 1.890 | 26.310 | 9.000 | 3.957 | 5.567 |
| Ours (Reported in the paper) | 5.879 | 1.988 | 21.240 | 9.698 | 3.471 | 5.647 |
| Ours (re-run-1)) | 5.896 | 1.944 | 20.016 | 9.807 | 3.473 | 5.803 |
| Ours (re-run-2)) | 6.015 | 1.987 | 20.456 | 10.291 | 3.420 | 5.699 |
| Ours(Mean±Variance) | 5.930±0.061 | 1.973±0.021 | 20.571±0.506 | 9.932±0.258 | 3.455±0.025 | 5.716±0.065 |

Table 12: Quantitative results of our method on FFHQ [35] and LSUN-Church [90], where error bars are reported in terms of mean and variance, and Ours is StyleGAN-V2+DMD.

|  | LSUN-Church (126K) | | FFHQ(70K) | |
|---|---|---|---|---|
|  | FID ↓ | IS ↑ | FID ↓ | IS ↑ |
| StyleGAN-V2 | 4.292 | 2.589 | 3.810 | 5.185 |
| Ours (Reported in the paper) | 3.061 | 2.792 | 3.299 | 5.204 |
| Ours (re-run-1)) | 3.025 | 2.795 | 3.177 | 5.225 |
| Ours (re-run-2)) | 2.993 | 2.787 | 3.285 | 5.200 |
| Ours(Mean±Variance) | 3.026±0.028 | 2.791±0.003 | 3.254±0.055 | 5.210±0.011 |

