# OpenReview forum: "Dynamically Masked Discriminator for GANs"
_NeurIPS.cc/2023/Conference — NeurIPS 2023 poster_

### Official Review · Reviewer_dcqD · 2023-06-13

**Soundness:** 4 excellent
**Presentation:** 4 excellent
**Contribution:** 4 excellent
**Rating:** 7
**Confidence:** 5

**Summary:**

This paper presents a novel method for training Generative Adversarial Networks (GANs) based on online continual learning, addressing the persistent challenge of GAN training instability.
The method considers the time-varying distribution of generated samples and prompts the discriminator to learn new information swiftly. It accomplishes this by detecting when the discriminator is lagging in learning from newly generated data and applying dynamic masking to discriminator features, thus forcing faster adaptation to the changing data distribution.
Improves GAN training, but also outperforms existing SOTA and advanced diffusion models.


**Strengths:**

1. Excellent pilot study to understand the generated distribution.
2. Method well explained. Nice flow chart
3. Promising image results supported by FID.
4. Multiple datasets


**Weaknesses:**

1. The video in the supplementary is confusing.
2. Lack of intuition explanation or theory proof to the proposed “Dynamically masked discriminator”
3. Fig3, 5 are both feature maps for visualization. I don’t understand the feature map, asking for more explanations.


**Questions:**

1. Does the proposed method introduce a significant amount of computation cost?
2. I can’t understand the feature maps used in the figure, asking for more explanation.
3 .How well does the proposed method generalize to other GAN methods?


**Limitations:**

Refer to previous sections.

---

> ### Author Rebuttal · Authors · 2023-08-10
>
> Thanks for the reviewer's positive feedback and insightful comments. We are delighted to learn that the reviewer agrees that " novel method", "excellent pilot study", "well explained", "nice flow chart", "promising image results", and "multiple datasets".
>
> **Q1. Asking for more explanations on the feature maps in Figs. 3,5**
>
> **A.**  Feature maps in Fig. 3 are extracted from the layer of StyleGAN-V2’s discriminator trained at different training steps.  These feature maps visualize the attentive regions of the discriminator, showing that StyleGAN-V2’s discriminator pays attention to almost fixed local regions over time, given the input image. That is, StyleGAN-V2’s discriminator almost depends on fixed local regions for discrimination.  However, as the generator evolves during training,  the discriminative local regions distinguishing between real and generated samples change over time.
> Hence, it is expected that the discriminator pays attention to different local regions in the samples at different steps, to learn the time-varying generated distributions(see Fig. 2).
>
> In Fig. 5, feature maps also show the attentive regions of the discriminator. For example, the feature map in the fifth column assigns high values to the right eye regions. By masking the feature map in the sixth column,  our method breaks the original dependency of the discriminator on some local features that are important to distinguish historical samples and encourage the discriminator to learn new knowledge from incoming data.
>
>
> **Q2: Intuition explanation or theory proof to the proposed “Dynamically masked discriminator**
>
> **A.** Thank you for the comments.  We agree with the reviewer that theory proof can make our work more comprehensive. We will explore it in future work, as it is beyond the scope of this paper.
> In this paper, we mainly show the challenges posed by the time-varying distributions, reveal that
> typical discriminators slow down their adaptation to the changes in the incoming data,
> and propose a new method to address the challenges.
>
> For the intuition explanation of our method,  as discussed in the responses to the above questions,  we aim to enable the discriminator to pay attention to different local regions for generated samples at different steps, such that the discriminator learns the time-varying distributions of generated samples.
> To this end, when the discriminator is detected to be retarded, our method dynamically masks the discriminator features to reduce its dependence on some local features that are important to distinguish historical samples (see Fig. 5). As a result,  the discriminator is enforced to re-build the dependency of incoming data and remaining non-masked local regions, and hence  learn new knowledge from the incoming data.
>
>
>
>
> **Q3: Does the proposed method introduce a significant amount of computation cost**
>
> **A.** Thanks for the comments.  Our method incurs a negligible increase in memory cost, since our method only additionally detects discriminator retardation and introduces feature masks.
>
> For computational cost, our method does not affect the inference time, since our method is designed for the discriminator and only the generator is used during inference. For training, our method introduces moderate time cost; however, this is affordable (e.g., increasing GPU number to reduce training time), and the additional training time cost depends on the time interval of estimating the retardation. Our method additionally introduces 6 hours 11 minutes for training on AFHQ in our experiments, when our method is integrated with StyleGAN-V2 and retardation is detected at every 4 kimgs. When detecting retardation at every 20 kimgs, our method only additionally takes 1.3 hours for training.
>
>
>
>
> **Q4:How well does the proposed method generalize to other GAN methods?**
>
> **A.** Thanks for the valuable comments. Following your suggestions, we evaluate the generalizability of our method to other GAN methods by incorporating our method with StyleGAN-V3 and unconditional BigGAN, respectively. Compared with StyleGAN-V3 using the original discriminator (FID  5.850),  our method improves the performance of StyleGAN-V3 by 15.88\% in the FID $\downarrow$ metric on AFHQ-Cat.  Similarly, our method effectively improves the training of unconditional BigGAN, reducing the FID  of unconditional BigGAN by 34.29\% on CIFAR10.

---

> > ### Comment · Reviewer_dcqD · 2023-08-18
> >
> > Thanks for the comments. I read them on day one. I will keep my original rating.
> >
> > Nice work!

---

> > > ### Author Response · Authors · 2023-08-18
> > > **Glad to Receive Your Response**
> > >
> > > Dear Reviewer dcqD,
> > >
> > > Thank you for your positive feedback and insightful review comments, which helped us significantly improve the paper's quality. We appreciate your time and effort in reviewing our work.
> > >
> > > Best,
> > >
> > > The Authors.

---

### Official Review · Reviewer_zVaP · 2023-07-02

**Soundness:** 3 good
**Presentation:** 3 good
**Contribution:** 2 fair
**Rating:** 5
**Confidence:** 4

**Summary:**

The paper introduces a new method to enhance GAN training. Taking a continual learning perspective, the authors contend that the discriminator faces challenges in modeling the real/fake distribution due to the dynamic nature of the generated samples' distribution over time. Consequently, this slows down the learning process of the discriminator. The paper suggests that this slowdown occurs because the discriminator is trained using historical data. To address this issue, the authors propose a method to detect the sluggishness of the discriminator and dynamically mask the feature map to accelerate its learning.

The authors perform experiments on several benchmark datasets and demonstrate that the proposed method surpasses several existing GAN methods, as measured by FID and IS scores. Additionally, they provide an ablation study to examine the optimal choice of the layer to be masked and the corresponding masking ratio for achieving the best performance.

In general, the paper is well-written and easy to understand. However, it should be noted that the proposed improvement is incremental, as the concepts of continual learning perspective and dynamic discriminator training have previously been introduced in [74]. The main contribution of this paper lies in the use of masking to adjust discriminator training. It is important to address some claims made in the paper that currently lack empirical evidence, and it would be beneficial to provide further ablation study on retard detection.

**Strengths:**

The proposed method introduces a technique to identify the slowdown and presents a novel way to mask feature maps to accelerate the discriminator learning.

The results are encouraging.

The paper is easy to follow.


**Weaknesses:**

I find the claim that the retardation of the discriminator is attributed to learning on historical data somewhat unclear. The slowdown of the discriminator could also be a result of overfitting, causing the gradient to vanish and making learning difficult. It is necessary to provide further justification for the claim that historical data is the cause of the retardation. I suggest the authors include an experimental study to support their argument.

I would suggest the author consider including a comparison of the mean and variance between the proposed method and StyleGAN-2 in Figure 2. This would provide valuable insights into how the generator distribution changes with the proposed method. Additionally, it would be helpful to have more details about the figure. For instance, it would be beneficial to know the number of samples used to calculate the mean and variance. Could the author clarify whether the plot represents early epochs or late epochs? If the figure displays early epochs, it would be intriguing to observe how the changes evolve in the later epochs.

Similarly, I recommend that the author consider showcasing the model parameter difference in Figure 3 to demonstrate that the learning of the proposed method does not slow down. This would provide valuable evidence to support the claim. Additionally, it would be helpful if the author could clarify where the retard detection takes place during the training process when the proposed method is applied in this example.

The detection of retardation is a crucial aspect of the proposed method and warrants an ablation study to assess its impact. It would be valuable if the authors could provide an ablation study that demonstrates how frequently retardation is detected during the training process of the proposed method. Additionally, it would be interesting to explore the consequences of removing retardation detection and using random masking consistently throughout the training.

Overall, the paper presents several intriguing aspects. However, it is crucial to provide justifications and supporting evidence for the claims made. Without proper justification, there is a risk of potentially misleading the research community. I would consider the paper to be on the borderline, and I am looking forward to reading the authors' rebuttal addressing my questions and the feedback from other reviewers to make a final decision.

**Questions:**

See questions and suggestions above.

---

> ### Author Rebuttal · Authors · 2023-08-10
>
> We appreciate the reviewer for the positive comments and
> constructive suggestions. The reviewer agrees that "a new method", "novel way to mask feature maps", "well-written and easy to understand",   "the results are encouraging", and "surpasses several existing GAN methods".
>
>
> **Q1: Experimental study to show that the retardation of the discriminator is attributed to learning on historical data.**
>
> **A.** Thank you for your suggestions. We conduct experimental studies inspired by the study in online continual learning for rapid adaptation [6].  Given the discriminator trained from the beginning to time $T$, we evaluate its historical knowledge retention by evaluating its discrimination performance on historical data generated at time $T$-$t$, and evaluate its rapid adaptation by evaluating its performance in current data at time $T$  and future data at  $T+t$.
>
> The table below shows that the discriminator of StyleGAN-V2 achieves high accuracy on historical data, however, performs much worse on current and future data. This shows that the discriminator of StyleGAN-V2  retains historical knowledge (i.e., high accuracy in historical data),  yet, does not fast learn and adapt to the changes in the distribution of future data.
>
> **Table**. The accuracy of the discriminator on historical data and future data, where the discriminator is trained at 800 kimgs step, and $(\cdot)$ indicates the data at time $T$-$t$ or $T+t$.
>
>
> |   Method |  Historical data (200)  | Historical data (600)    | Current data (800)    | Future data (1000)|
> |:-------:|:----------:|:----------:|:----------:|:----------:|
> |StyleGAN-V2|0.9696| 0.9488|0.8236|0.604  |
> |Ours (StyleGAN-V2 + DMD)|0.9129 |0.9229|0.8793|0.6543|
>
> Instead, by detecting retardation and masking the discriminator, our method enforces the discriminator to reduce retained historical knowledge (see decreased accuracy on historical data), while effectively encouraging it to fast learn and adapt to new knowledge of future data (i.e., our method achieves higher accuracy on current and future data than StyleGAN-V2).
>
> In addition, we calculate the average gradient of the StyleGAN-V2 during the training phase, showing the gradient does not vanish from 0 to 1200 kimgs.
>
> **Table**: Average gradient of the StyleGAN-V2  during training
>
> | kimgs  |  Gradient |
> |:-----:|:------:|
> | 0  | 3.5145e-08   |
> | 200  | 4.6110e-07   |
> | 400  | -9.4235e-07   |
> | 600  | 5.6989e-07   |
> | 800  | -4.3328e-07   |
> | 1000  | 9.2649e-08   |
> | 1200  | 1.2138e-06   |
>
>
>
>
>
>
>  **Q2: The number of samples used to calculate the mean and variance in Figure 2.**
>
> **A.** We use 50k samples to calculate each mean and variance values.
>
>
> **Q3: Whether the plot in Figure 2 represents early epochs or late epochs?**
>
>  **A.** The plot in Figure 2 represents the whole training process from early epochs to late ones on FFHQ.
>
>
>
> **Q4: Model parameter difference of the proposed method and where the retard detection takes place.**
>
>
> **A.** Thanks for your valuable suggestions. Our method detects the discriminator to be retarded at 1000 kimgs after 800 kimgs.  Our method then applies the proposed dynamic discriminator adjustment to train StyleGAN-V2 from 1000 to 1004 kimgs, which increases the model difference of our method by 63.889\%, compared with the model difference between 800 to 1000 kimgs. Instead, the model difference of StyleGAN-V2's discriminator is decreased by 2.778\% without our method. These results show StyleGAN-V2's discriminator slows down learning and our method effectively encourages the discriminator to fast learn.
> We will add our model's parameter difference in Fig. 3, as suggested.
>
>
> **Q5: How frequently retardation is detected during the training process of the proposed method.**
>
> **A.** The number of detected retardation is 256 times in total on AFHQ-Cat，when StyleGAN-V2 + DMD (ours) is trained for 1400 kimgs steps.
>
>
> **Q6: Removing retardation detection and using random masking consistently throughout the training.**
>
> **A.**  Thank you for the suggestions. We build a baseline, namely Random-Mask Discriminator (RMD), which removes retardation detection from our method and uses random masking consistently throughout the training. We also build the second baseline,  namely Time-Varying-mask Discriminator (TVD), which removes retardation detection and randomly changes the mask per mini-batch.
>
> The table below shows that our method outperforms RMD and TVD by a large margin, thanks to the proposed retardation detection and dynamic discriminator adjustment modules.
>
> **Table**: Comparison with the different masking strategies on AFHQ-Cat (256 $\times$ 256 pixels).
>
> |  Method |  FID $\downarrow$ |
> |:---------------:|:-----:|
> |  StyleGAN-V2   | 7.924 |
>   StyleGANV2 + TVD       | 9.294
> | StyleGAN-V2 + RMD      | 8.759 |
> |   Ours (StyleGANV2+ DMD)    | **5.879** |

---

> > ### Author Response · Authors · 2023-08-21
> >
> > Since the author-reviewer discussion is getting closing soon,  we believe we have addressed your questions and comment, and we hope to hear from you.  Thank you for your kind consideration!
> >
> > **Q. Difference from DynamicD [74]**
> >
> > **A.** Our paper is inspired by the insightful observation in DynamicD [74] and appreciates its valuable observations, as stated in Lines 35-36 of the main paper. However, our method and DynamicD [74] are different in three aspects:
> >
> > •	 DynamicD [74]  focuses on a problem different from ours. Our method focuses on detecting the retardation of the discriminator and encouraging the discriminator to rapidly adapt to new data, which haven’t been explicitly considered by existing GANs methods. Differently, DynamicD [74] focuses on adjusting the discriminator model capacity according to various data scales: increasing the model capacity for large training data, while decreasing the layer width given limited data.
> >
> > •	DynamicD [74] does not mention continual learning, while our method proposes a new perspective, i.e.,  online continual learning to address GAN training.
> >
> > •	Although our method is fully automated,  our method outperforms DynamicD [74] by a large margin (20.93%) on LSUN-Church (Table 2 in the main paper).

---

### Official Review · Reviewer_J72A · 2023-07-05

**Soundness:** 2 fair
**Presentation:** 3 good
**Contribution:** 2 fair
**Rating:** 4
**Confidence:** 4

**Summary:**

This paper proposes a new training method of generating adversarial networks from the perspective of online continuous learning. In order to address the challenges posed to the discriminator by the dynamic changes in the generated data distribution during training, the authors propose to detect whether the discriminator has slowed down its adaptation to newly generated data and dynamically mask its features to force it to quickly learn new knowledge. Experimental results on FFHQ, AFHQ, and LSUN-Church show that the proposed method is superior to the state-of-the-art methods in image generation.

**Strengths:**

- Overall, the paper is well-organized and well-written, making it easy to follow. The analysis of the discriminator in the Pilot Study section is helpful for readers to understand the motivation of the work.
- The paper provides a comprehensive review of related work, which helps readers to better understand its contributions.
- The proposed method outperforms state-of-the-art methods, including diffusion models, in mainstream experiments according to the experimental results reported in the paper.

**Weaknesses:**

- The notation used in the method description is confusing and inconsistent. For example, $\mathcal{D}$ and $\bar{D}$, $\theta^t$ and $\phi(t)$. Additionally, $\phi(t)$ is not included in discriminators as a parameter like $\theta^t$.
- The comparison of the visualization results is not significant. For instance, it is difficult to understand the improvements in the generated results of the proposed method compared to the baseline in Fig.1. Considering that this is a subjective evaluation, including objective evaluation metrics in this experiment would be helpful.
- The proposed method is kind of heuristic in nature, hence the technical novelty and contribution are limited. More advanced methods for online continual learning may be a better choice.

**Questions:**

- Considering that the proposed method requires continual detection of discriminator retardation, will this result in significant additional training overheads compared to baselines?

**Limitations:**

The authors have not adequately addressed the **limitations of the proposed method** and potential negative societal impact of their work.

---

> ### Author Rebuttal · Authors · 2023-08-10
>
> We appreciate the reviewers for the careful reviews and
> constructive suggestions.  We are encouraged that the reviewer agrees that our paper is "well-organized and well-written", "new training method ", "provides a comprehensive review", "Pilot Study section is helpful for readers to understand the motivation", and  "outperforms state-of-the-art methods".
>
> **Q1: The notation used in the method description.**
>
> **A.** Thank you for pointing this out. We will replace $\bar{D}$ by $\bar{\mathcal{D}}$ to maintain notation consistency between the discriminator without masks and that with masks.  We will include $\phi(t)$ in the discriminator like $\theta^t$ in the final version.
>
> **Q2. Including objective evaluation metrics in Figure 1**
>
> Thank you for your constructive suggestions. Firstly, we introduce FID $\downarrow$ to evaluate generation quality. The FID of our method is 3.299, suppressing the baseline (FID 3.810).
>
> Secondly, we introduce the cosine similarity metric. We aim to enable the discriminator to pay attention to different local regions in samples at different steps, such that the discriminator adapts to the time-varying distributions of generated samples. We extract binary attention maps from the feature maps in the second row of Figure 1, and then calculate the cosine similarity of the attention maps between the current training step and the previous one. Lower value indicates better performance, i.e., attentive regions at the current step are more different from that at the previous one. The table below shows our method enforces the attentive regions of the discriminator to be more different at each training step than the baseline, better adapting to time-varying distribution of generated data. We will include objective evaluation results in the final version.
>
>
> **Table 2**: Cosine similarity $\downarrow$ of attention maps between current and previous steps in Figure 1.
>
> |          |   t1  | t2   |   t3  |
> |:--------:|:------:|:------:|:------:|
> | Baseline | 0.8672 | 0.8242 | 0.8145 |
> |   Ours   | **0.2871** | **0.4336** | **0.5488** |
>
>
>
>
>
> **Q3: More advanced methods for online continual learning.**
>
> **A.** We will explore more advanced methods in our future work, thank you for your suggestion. Yet, as recognized by Reviewer zHGZ, "Rather than designing a complex network architecture, their method is designed to be easily integrated into any existing discriminator or used in combination with data augmentation methods which increases the significance and broader impact of the work".  Reviewer dcqD also agreed that we propose "a novel method for training Generative Adversarial Networks (GANs) based on online continual learning" which achieves "Promising image results" on "Multiple datasets".
>
> We design such a simple and straightforward method, due to the specific challenges introduced by GAN models.  Different from existing online continual learning methods for rapid adaption (e.g., for classification), GANs models posed new challenges in that the training GANs plays a min-max game between the generator and the discriminator. Sophisticated designs may increase the difficulties of optimizing the generator and the discriminator, leading to large instabilities in the training.
> Different from [74] which explores the influence of various capacities of the discriminator on training,
>  we propose to detect the retardation of the discriminator and dynamically mask the discriminator to force it to learn fast, which is simple yet effective.
>
> Nevertheless,  we thank the reviewer's suggestions.  We try to improve our discriminator retardation detection.  In particular, if the Retardation value $\mathcal{R}_t$ of current time interval $t$ is higher than that of the previous one, the discriminator would be detected in retardation. With this new discriminator retardation detection, our method achieves  5.877 FID on AFHQ-Cat(256 $\times$ 256), which is on par with ours (5.879 FID) in the main text.
>
>
>
> **Q4: Will the detection of discriminator retardation result in significant additional training overheads**
>
> **A.** Our method does not result in significant additional training overheads. In particular, the retardation of the discriminator is typically not an abrupt, but smooth transition process during training. This enables our method to detect retardation of the discriminator sparsely, instead of after each training image.  The additional time cost is 6 hours 11 minutes for the whole training on AFHQ in our experiments,  where our method is integrated with  StyleGAN-V2 and detects retardation at every 4 kimgs. When detecting at every  20k kimgs, our method additionally takes 1.3 hours.
>
> **Q5: Limitations of the proposed method and potential negative societal impact**
>
> **A.** Thank you for the comments.  Due to the page limitation, we discuss the limitation of the proposed method in Sec. 5 of the Appendix:
>
> Theoretical studies can make this work more comprehensive; however, we have not explored it in the paper, since it is beyond the scope of this paper. Moreover, while the proposed method can effectively improve the training of the CNN-based GANs models, combining our method with Transformer-based ones is left to be investigated in the future.
>
> Potential negative societal impact is also discussed in Sec. 5 of the Appendix:
>
> Like other generative models, our method can be misused for applications, such as Deepfake [1], where fake content is synthesized to deceive and mislead people, leading to a negative social impact. Nevertheless, many researchers have considered this problem, while exploring fake content detection and media forensics techniques.

---

### Official Review · Reviewer_ZBcx · 2023-07-07

**Soundness:** 4 excellent
**Presentation:** 4 excellent
**Contribution:** 3 good
**Rating:** 4
**Confidence:** 4

**Summary:**

In this paper, the authors propose a novel perspective for generative adversarial training via continual learning, which achieves better results on the generative quality for local details and better quantitative metrics. To realize continual learning, the authors propose two models, 1) the discriminator slow-down detection mechanism and 2) the dynamical masked discriminator for different time-steps of training.

**Strengths:**

1. The novel perspective of modelling GAN with continual learning is interesting and promising, which tries to improve GAN training on discriminators with fast adapting features.
2. The proposed method is simple and straightforward yet effective to be trained and probably would be general for any type of GAN network.
3. The model is well-explained, and the paper is well-written.

**Weaknesses:**

1. Even though the idea of continual learning for GAN is novel and exciting, it seems that the connection between continual learning and the dynamically masked discriminator is not well-explained.
2. It seems the improvements on the face dataset or animal face dataset shows marginal improvement; I would be more interested in the training on a more diverse dataset as the SOTA generative models are mostly capable of generating on large-scale datasets, such as imagenet1000.

**Questions:**

Please refer to the above weakness section.

**Limitations:**

The limitation is not discussed, however I would concern if the method can be directly scaled up to the large scale dataset generation, such  as imagenet1000.

---

> ### Author Rebuttal · Authors · 2023-08-10
>
> We appreciate the reviewer's valuable comments. We are encouraged that the reviewer agrees that our work is “interesting and promising”, “novel and exciting”, “simple and straightforward yet effective to be trained”, “probably would be general”, “well-explained”, and “well-written”.
>
> **Q1.The connection between continual learning and the dynamically masked discriminator.**
>
> **A.** Thank you for the comments. Our method is inspired by the recent work on online continual learning for rapid adaptation. The study pointed out that data with time-vary distribution requires learning methods to have the ability of fast learning and quick adaption (see Line 47-55 in the main paper). For GANs, the distribution of the generated data drifts over time, as the generator evolves during training. This also requires the discriminator to fast learn new knowledge of incoming data.
>
> However, online continual learning for rapid adaption has been underexplored  [16, 6, 39]. Moreover, different from most online continual learning methods for rapid adaption, training GANs models plays a min-max game between the generator and the discriminator, posing new challenges.  For example,  the incoming data of the discriminator is unknown and uncertain, while the generation results of the generator depend on the discriminator and the discriminator learns from data generated by the generator.  Yet, these challenges haven't been considered by existing online continual learning methods.
> To address the challenges, we propose a new method for GAN models, which detects the discriminator retardation and masks the discriminator features. More specifically, masking the feature map of the discriminator is to break the original dependency of the discriminator on some local features that are important to distinguish historical data, and enforce the discriminator learn new knowledge of incoming data by rebuilding the dependency of remaining non-masked local features and incoming data.
>
>
>
> **Q2: Scale up to the large-scale dataset generation, such as ImageNet-1000.**
>
> **A.** Thanks for the valuable comments. Following your suggestions, we train our method on ImageNet-1000. Yet, the settings of SOTA GAN methods typically require a larger number of iterations (e.g., 150,000 iterations for BigGAN [5]) on ImageNet-1000,  which takes more than two weeks to train from scratch with eight NVIDIA V100 GPUs [5]. Due to the time limitations and limited number of GPUs, we integrate our method with an intermediate checkpoint of the BigGAN model which has been trained for 100,000 iterations.  We then further trained models from 100,000 to 107,500 iterations on ImageNet-1000 at 128$\times$128  resolution using two NVIDIA V100 GPUs.
>
> Compared with the original BigGAN that achieves the FID $\downarrow$ of 12.213 at the 107,500$^{th}$ iteration, our method (i.e., DMD) improves the training of BigGAN, outperforming the original BigGAN by **8.196\%** (see the table below). This not only shows that our method benefits the training of BigGAN on large-scale datasets, but also demonstrates the flexibility and compatibility of our method in combination with GAN models such as pre-trained GAN models.
>
> **Table**:  Results of the proposed method on  ImageNet-1000.
>
> |                 Method                |  FID $\downarrow$ |
> |:-------------------------------------:|:-----:|
> |           BigGAN           | 12.213|
> |     Ours (BigGAN+ DMD)    |**11.212** |
>
>
> In addition,  Table 2 in the main paper evaluates our method on a large dataset LSUN-Church, which contains 126K images and is used to challenge generative methods in recent GAN and diffusion work (e.g., [54,74]). Our method outperforms StyleGAN-V2 by 28.7\% on LSUN-Church, and surpasses a SOTA diffusion model LDM [54] by 23.9\%.
>
>
> **Q3:  The limitation.**
>
> **A.** Thanks for your suggestions. Due to the page limitation, we have to discuss the limitation of our method in  Sec. 5 of the appendix:
>
> Theoretical studies can make this work more comprehensive; however, we have not explored it in the paper, since it is beyond the scope of this study. Moreover, while the proposed method can effectively improve the training of the CNN-based GANs models, combining our method with Transformer-based ones is left to be investigated in the future.

---

> > ### Comment · Reviewer_ZBcx · 2023-08-20
> >
> > thanks for the reply, my questions are partially addressed. But, I am still not convinced by the continual learning with the dynamically masked strategy. Here is the question for " treating generated samples throughout training as a stream of data slows down learning of D",
> > the intuition is still not clear here. I got the point that D would fall into some lazy easy pattern, and masked strategy can force them focus on some more detailed and harder features for discrimination. I am still doubt why this would connect to continual learning, which is a concept to learn a model for a large number of tasks sequentially without forgetting knowledge obtained from the preceding tasks.

---

> > > ### Author Response · Authors · 2023-08-21
> > >
> > > Thank you for your time and effort in evaluating our work.
> > >
> > > **Q1. Why this would connect to continual learning, which is a concept to learn a model for a large number of tasks sequentially without forgetting knowledge**
> > >
> > > **A.** Thank you for the comments. Our work is in line with the emerging research direction: online continual learning towards rapid adaptation. Continual learning can be classified into two categories: offline and online [16], where offline continual learning mainly aims to mitigate forgetting. Recent online continual learning methods [16, 6, 39]  focus on enabling **rapid adaptation to new incoming data**, where method [6] has revealed that *information retention (mitigate forgetting) and rapid adaption are conflicting objectives, requiring careful compromises*.
> > >
> > > In other words,  compared with offline continual learning, both methods [16, 6, 39] and our method address a more challenging but realistic problem. That is,  (1) training data is seen only once;  (2) the data is **not independent and identically distributed**; and (3) the distribution of data under **a fast-changing distribution shift**. This drives [16, 6, 39] and our method to focus on rapid adaption.
> > > Reviewer dcqD agrees that we propose "a novel method for training Generative Adversarial Networks (GANs) based on online continual learning, addressing the persistent challenge of GAN training instability."
> > >
> > > In addition, one of the most recent continual learning works [R1] also points out that it is unnecessary to remember all historical information, and "forgetting non-recurring information is not catastrophic”, which is in line with our work. We will clarify it in our final version.
> > >
> > > [R1] Saurabh Kumar,  Henrik Marklund, Ashish Rao, Yifan Zhu, Hong Jun Jeon, Yueyang Liu, and Benjamin Van Roy. Continual Learning as Computationally Constrained Reinforcement Learning. arXiv preprint arXiv:2307.04345, 2023.
> > >
> > > **Q2. The question for "treating generated samples throughout training as a stream of data slows down learning of D"**
> > >
> > > **A.**  Our manuscript does not have such a statement. We guess this refers to "We propose to detect whether the discriminator slows down the learning of new knowledge in generated data by treating generated samples throughout training as a stream of data" (see L51 in the main paper).  This does not mean that treating generated samples as a stream of data slows down the learning of D. Instead, it means that we treat generated samples as a stream, enabling us to detect whether the discriminator slows down learning new knowledge.
> > >
> > > We will revise it and make it clear.

---

### Official Review · Reviewer_zHGZ · 2023-07-07

**Soundness:** 3 good
**Presentation:** 3 good
**Contribution:** 3 good
**Rating:** 7
**Confidence:** 4

**Summary:**

The paper proposes Dynamically Masked Discriminator (DMD) which automatically adjusts the discriminator by dynamically masking the features when learning slows down, forcing the discriminator to learn new knowledge in the generated data. It consists of two modules: (1) discriminator retardation detection and (2) dynamic discriminator adjustment. The first module detects when the discriminator starts to learn slower (i.e. rely on old knowledge rather than learn new distributions of generated data). The second module dynamically assigns or removes masks to the features of the discriminator. The experiment results show that their method helps increase the quality of generated samples across diverse model architectures and datasets.

**Strengths:**

* The paper is well-motivated and clearly presented.
* Rather than designing a complex network architecture, their method is designed to be easily integrated into any existing discriminator or used in combination with data augmentation methods which increases the significance and broader impact of the work.
* They perform extensive experiments on a wide range of datasets and baseline architectures against various existing methods that improve discriminator training. Experiment results show that their method significantly reduces the FID score when combined with state-of-the-art GAN architectures. Qualitative results demonstrate that the method helps generate higher-quality samples without artifacts.
* The paper is clearly organized and provided with sufficient technical and implementation details.


**Weaknesses:**

* Masking seems to be an important design factor for the efficacy of this method. The authors should elaborate on why switching from non-mask training to mask training is more desirable than continuously changing the mask ratio over time.
* There exists minor typos and grammar mistakes throughout the paper.
* The paper lacks the limitation section, which I feel would improve the paper.

**Questions:**

* What is the benefit of strictly separating the non-mask and mask training stage rather than changing the mask ratio over time?
* Has the authors experimented with higher-resolution images? This would increase the impact of this work.

**Limitations:**

The authors do not mention the limitation of their work.

---

> ### Author Rebuttal · Authors · 2023-08-10
>
> We would like to thank the reviewer for their insightful feedback and constructive comments. We are encouraged by the reviewer's positive comments, such as “well-motivated”, “clearly organized”, “easily integrated into any existing discriminator”, “significance and broader impact of the work”, "extensive experiments", "improve discriminator training", and "significantly reduces the FID score".
>
> **Q1: Comparison with changing the mask ratio over time.**
>
> **A**. Thanks for the valuable comments. Following your suggestion, we build a baseline, namely Continualy-Changed-mask-ratio Discriminator (CCD), which replaces our dynamic discriminator adjustment module by gradually increasing the mask ratio from 0.1 to 0.9 or decreasing the mask ratio from 0.9 to 0.1 over time.
>
> The table below shows that our method outperforms CCD, since CCD increases instabilities in GAN training. More specifically, the process of GAN training is to play a min-max two-player game between the generator and discriminator, which is more unstable than typical classification problems. Compared with our method,  CCD causes the discriminator's discrimination ability to be more frequently changed and uncertain, making it more difficult for the generators to fool the discriminator. The deteriorated performance of the generator would further negatively affects the training of the discriminator. Instead, when the improvement of the discriminator slows down, our method switches from non-mask training to mask training, otherwise maintains masking/non-masking at a time interval, which provides more stable training than CCD.
>
>
>  **Table**: Comparison with changing the mask ratio over time on AFHQ-Cat (256$\times$256 pixels).
>
> |                 Method                |  FID $\downarrow$ |
> |:-------------------------------------:|:-----:|
> |            StyleGAN-V2             | 7.924 |
> |         StyleGAN-V2 + CCD   | 8.441 |
> |     Ours (StyleGAN-V2+ DMD)    | **5.879** |
>
>
> In addition, we observe that continually changing the mask ratio requires careful designs on the decay/growth strategy of the mask ratio, and needs to enforce it to be adaptive to retardation detection. We thank the reviewer's comments and think temporally changing the ratio can be a promising direction.  We will explore this further in future work.
>
>
>
> **Q2: Minor typos and grammar mistakes.**
>
> **A.** Thank you. We will correct typos and grammar mistakes.
>
> **Q3: Has the authors experimented with higher-resolution images?**
>
> **A.** Thank you for the constructive comments. We conduct experiments on the high-resolution AFHQ-Cat dataset, where the resolution of an image is 512 $\times$ 512 pixels. By integrating our method with the SOTA method StyleGAN-V2,  our method reduces the FID $\downarrow$ of StyleGAN-V2 by a margin of **18.77\%** (from 4.3160  to 3.5061) on AFHQ-Cat (512 $\times$ 512), effectively improving the performance of StyleGAN-V2 on higher-resolution images.
>
> **Q4: The limitation section.**
>
> **A.** Thanks for your suggestions. Due to the page limitation, we have to discuss the limitation of our method in  Sec. 5 of the appendix:
>
> Theoretical studies can make this work more comprehensive; however, we have not explored it in the paper, since it is beyond the scope of this study. Moreover, while the proposed method can effectively improve the training of the CNN-based GANs models, combining our method with Transformer-based ones is left to be investigated in the future.

---

> > ### Comment · Reviewer_zHGZ · 2023-08-18
> >
> > Dear authors,
> >
> > Thank you for your detailed response. The authors have addressed all of my concerns. After reading comments from other reviewers and your feedback, I plan to keep my score.

---

> > > ### Author Response · Authors · 2023-08-18
> > > **Thank you for your valuable and positive feedback!**
> > >
> > > Dear Reviewer zHGZ,
> > >
> > > We sincerely thank the reviewer for your insightful comments and positive feedback.  Your constructive review comments help us significantly improve the quality of the manuscript and enhance the strength of our work.
> > >
> > > Best regards,
> > >
> > > The Authors.

---

### Author Response · Authors · 2023-08-21
**Thank ACs and all Reviewers and a Rebuttal Phase Recap**

We would like to thank ACs and all reviewers for your time and effort in evaluating our paper and facilitating the discussion.

As the author-reviewer discussion is closing soon, we would like to briefly summarize the discussion status and outcomes.

1. Reviewers [zHGZ](https://openreview.net/forum?id=sodl2c3aTM&noteId=bkFJawnnMc), and [dcqD](https://openreview.net/forum?id=sodl2c3aTM&noteId=0W1gH4GkHJ) engaged proactively in the discussion phase. Both Reviewers zHGZ and dcqD **maintained a rating of 7**, Reviewer zHGZ states that we have addressed all of his/her concerns, and  Reviewer dcqD agrees that our paper is [**“nice work”**](https://openreview.net/forum?id=sodl2c3aTM&noteId=JDTpx9D3nN).

2. Reviewer [J72A](https://openreview.net/forum?id=sodl2c3aTM&noteId=vTzVfakz3R)  does not reply during the discussion phase. His/her main concerns are two-fold: i) additional training overheads and ii) technical novelty and contribution. For i), Reviewer dcqD also raised a [similar concern](https://openreview.net/forum?id=sodl2c3aTM&noteId=0W1gH4GkHJ) initially. However, *Reviewer dcqD has confirmed that our responses have addressed this concern*.
 For ii), we have carefully explained why we design such a simple yet effective method. We only use the heuristic Retardation metric in the discriminator retardation, while proposing a new method (pipeline)  and dynamic discriminator adjustment. As recognized by  Reviewer  dcqD, we ["present a **novel method** for training Generative Adversarial Networks (GANs) based on online continual learning"](https://openreview.net/forum?id=sodl2c3aTM&noteId=0W1gH4GkHJ), and Reviewer zHGZ agrees that  ["Rather than designing a complex network architecture, their method is designed to be **easily integrated into any existing discriminator** or used in combination with data augmentation methods which increases the significance and broader impact of the work."](https://openreview.net/forum?id=sodl2c3aTM&noteId=bkFJawnnMc)  Accordingly, we are hereby confident that we've addressed Reviewer J72A's concerns, and we are optimistic about Reviewer J72A's feedback.

3. Reviewer [zVaP](https://openreview.net/forum?id=sodl2c3aTM&noteId=V4h3jQri7z) does not engage in the discussion phase. Yet, we value Reviewer's main suggestions: provide justifications and supporting evidence.   As suggested, we have [**conducted various studies and experiments**](https://openreview.net/forum?id=sodl2c3aTM&noteId=NqgP8Zerlh), including i) relying on historical data leads to the discriminator retardation;  ii) studying the average gradient of the StyleGAN-V2; iii) showcasing the model parameter difference, and demonstrating the learning of our method does not slow down; iv) how frequently retardation is detected; v)  exploring the results of removing retardation detection and using random masking consistently. The results showcase the contributions of our method in detecting discriminator retardation and enabling fast adaption. We believe that we've addressed Reviewer zVaP's concerns, and we are still looking forward to Reviewer zVaP's feedback.

4. Reviewer [ZBcx](https://openreview.net/forum?id=sodl2c3aTM&noteId=iHAR8bgLu0) still has a concern about the connection between continual learning and our work, since [*"continual learning is a concept to learn a model for a large number of tasks sequentially without forgetting knowledge"*](https://openreview.net/forum?id=sodl2c3aTM&noteId=XBlCXQShSB). Yet, rapid adaption and information retention are two main concepts in online continual learning. As listed in [our response](https://openreview.net/forum?id=sodl2c3aTM&noteId=9klE3vWgiM),  online continual learning towards rapid adaption [16, 6, 39]  focuses on fast adapting to new data, which is in line with our work.   Reviewer dcqD confirms this connection.  We suppose that Reviewer  ZBcx's concerns are addressed, and we are still eagerly awaiting Reviewer  ZBcx's feedback.

We sincerely appreciate the insightful comments and constructive suggestions of all reviewers that help us improve our work a lot!  We remain firmly committed to incorporating additional experimental results in our final paper to improve our work.

---

### Decision · Program_Chairs · 2023-09-21

**Decision:**

Accept (poster)

**Comment:**

The paper studies the use of online continual learning techniques when training generative adversarial nets (GAN), which addresses the issue that the discriminator faces a changing distribution during GAN training. The paper received a bi-modal assessment. Two reviewers were excited, two more concerned and one unsure. Major concerns were related to additional explanations (why switching from masking to non-masking and the use of masking in general), technical novelty, confusing notation, among others. The rebuttal was able to address most of the concerns. One reviewer remained concerned, missing a more theoretical explanation while the other reviewer didn't respond. AC thinks the paper presents an interesting aspect for GAN training that's useful for the community to discuss. AC also thinks that some choices (e.g., masking) could be better motivated and studied (at least empirically). Overall benefits outweigh concerns.